# Fetal Ovarian Cyst—A Scoping Review of the Data from the Last 10 Years

**DOI:** 10.3390/medicina59020186

**Published:** 2023-01-17

**Authors:** Carmen Bucuri, Dan Mihu, Andrei Malutan, Valentin Oprea, Costin Berceanu, Ionel Nati, Maria Rada, Cristina Ormindean, Ligia Blaga, Razvan Ciortea

**Affiliations:** 1Dr. Constantin Papilian Emergency Military Hospital, “Iuliu Hatieganu” University of Medicine and Pharmacy, 400610 Cluj-Napoca, Romania; 22nd Department of Obstetric & Ginecology, “Iuliu Hatieganu” University of Medicine and Pharmacy, 400610 Cluj-Napoca, Romania; 3Department of Obstetrics and Gynecology, University of Medicine and Pharmacy of Craiova, Emergency County Hospital of Craiova, 200051 Craiova, Romania; 42nd Department of Neonatology, “Iuliu Hatieganu” University of Medicine and Pharmacy, 400610 Cluj-Napoca, Romania

**Keywords:** fetal ovarian cyst, ultrasound, Nussbaum criteria

## Abstract

Abdominal cystic masses are diagnosed during the intrauterine period and have a relatively low incidence. Fetal ovarian cysts are the most common form diagnosed prenatally or immediately after birth. The pathophysiology of the development of these types of tumors is not fully elucidated, with ovarian hyperstimulation caused by maternal and placental hormones being the most accepted hypothesis. During intrauterine development, the diagnosis of fetal ovarian cysts is most often made accidentally during usual check-up ultrasounds corresponding to the first, second, and third trimesters of pregnancy. We conducted a scoping review with the aim to map the current knowledge regarding the treatment of fetal ovarian cysts diagnosed in the intrauterine period. Focusing on the articles published in the last 10 years in the specialized literature, we tried to identify a conceptualization regarding the surveillance and treatment of these anomalies.

## 1. Introduction

Abdominal cystic masses diagnosed during the intrauterine period and with a relatively low incidence represent a pathology with major importance due to the negative impact they can have on the health of the newborn and the future woman. The origin of these tumors can be any intra-abdominal organ, and thus the differential diagnosis must be made between ovarian, hepatic, pancreatic, and splenic cysts; intestinal duplications; choledochal cyst; meconial pseudocyst; renal cyst; and other cystic tumors. Fetal ovarian cysts are the most common form diagnosed prenatally or immediately after birth, with a reported incidence of 1:2500 births [1,2,3].

The pathophysiology of the development of these types of tumors is not fully elucidated, with ovarian hyperstimulation caused by maternal and placental hormones being the most accepted hypothesis. Thus, their development and detection take place in the third trimester of pregnancy, after 28 weeks of gestation, when the fetal ovary can respond to the stimulation produced by the hormonal status expressed in excess. Another hypothesis is that proposed by Enríquez et al., who in their paper describe in four cases the high position in the abdomen of these cystic masses, corresponding to the embryonic stage of 5–6 weeks of gestation [4]. This hypothesis proposes the vascular compromise as the cause of the complex ovarian cysts in the embryonic period; in the described cases, no ovarian tissue was identified. Moreover, a link between the incidence of these tumors and pregnancy complications such as pre-eclampsia, isoimmunization, or maternal diabetes has been demonstrated [3,5].

During the intrauterine development, the diagnosis of fetal ovarian cysts is most often made accidentally during usual check-up ultrasounds corresponding to the first, second, and third trimesters of pregnancy. Ultrasonography is the most useful tool in this case. If the diagnosis is unclear, or if the association of other fetal structural changes is suspected, it will be recommended that a fetal MRI be performed with the aim of a correct diagnosis as early as possible. Sometimes, the presence of ovarian cysts is detected only at the time of birth, accidentally, or because of symptoms caused by them. The most frequent symptoms encountered are represented by the compression exerted by the presence of the intra-abdominal mass, of which the most severe form is respiratory distress. This is as a result of pulmonary hypoplasia caused by the elevation of the diaphragm. Even asymptomatic ovarian cysts present an increased risk of complications such as torsion, intracystic hemorrhage, or bursting with hemoperitoneum that require an urgent surgical procedure.

According to a series of ultrasonographically assessed parameters, Nussbaum et al. classified ovarian cysts as simple and individually complex [6]. Simple ovarian cysts are described as unilocular, having a regular outline, with a thin wall, being well demarcated, with transonic intracystic content. In contrast, complex ovarian cysts have a thickened wall and their content is inhomogeneous due to the presence of intracystic hemorrhage, septa, or vegetations [6]. Complex types present a higher risk of complications and are more frequently symptomatic. Another extremely important ultrasonographic parameter, before establishing the therapeutic behavior, is their size. Most authors establish a cut-off of 20 mm as the minimum size for ovarian cyst diagnosis, with below this size being identified as the ovarian follicle [7]. Specific to ovarian cysts, the presence of a daughter cyst—a small, round structure with transonic content—has been described inside them [8,9]. Ovarian cysts rarely exceed 100 mm in size. The risk of complications and symptoms are directly proportional to their size, which is why many authors have proposed dividing them into “small” and “large”, with the cut-off between the two categories being 40 mm [2].

Regarding the recommended therapeutic conduct, there is no established international consensus. The treatment possibilities are represented by the “watch and wait” conservative approach, intrauterine aspiration, and postnatal surgical treatment. The choice of the treatment method is made, taking into account the type of cyst (simple or complex); their size; the presence of symptoms or complications; and, of course, the wishes of the family. Thus, simple, small ovarian cysts that do not present clinical symptoms and are not complicated are the perfect candidates for surveillance. Aspiration of the intrauterine cyst is recommended by most authors for simple cysts of increased size that, through the compression exerted, may affect the development of other organs or may become complicated. The presence of complications of clinical symptoms, especially regarding complex cysts of increased size, require surgical treatment. Many authors propose surgical treatment when the size of the cyst is over 40 mm, even in the absence of symptoms due to the increased risk of complications [7,10].

Supervision during intrauterine fetal development is recommended to be done at an interval of 2 weeks. The evolution of the size of the cyst and the change in the characteristics is monitored ultrasonographically. The torsion of the cyst and implicitly the ipsilateral adnexa is recognized as being among the most severe complications due to the frequent loss of viability of the ovary. The early detection of torsion is analyzed by Doppler analysis of the vascularization at the level of the affected adnexa. However, technological limitations predispose us to a high rate of false negative results.

The applied surgical treatment can be classified into two categories, depending on the possibility of preserving the ovarian tissue or not. Thus, surgical techniques that preserve the ovarian tissue include ovarian cystectomy and cyst aspiration. Oophorectomy and respective salpingo-oophorectomy are radical surgical techniques that do not preserve the ovarian tissue. Another situation encountered is the absence of ovarian tissue caused by a previous torsion, in which case that ovary is self-amputated. The surgical treatment when it is applied must always aim to save the ovarian tissue, and the implications of losing this reserve may have a major negative impact on the reproductive future of the fetus in question. From a technical point of view, interventions can be performed classically or laparoscopically.

## 2. Materials and Methods

The protocol of the present study was carried out on the basis of the model proposed by Arskey and O’Malley [11] and revised by the Joanna Briggs Institute [12]. The main objective is to synthesize the results of applying conservative and surgical treatment in the selected studies. In order to obtain a homogeneity of our study, only “case series”-type articles were included, with “case report”-type publications, systematic reviews, and meta-analyses being excluded. The steps taken in this regard were as follows: identification of the main objective -> identification of relevant studies -> selection of studies -> extraction of data -> presentation of results.

The objective of this scoping review was to map the current knowledge regarding the treatment of fetal ovarian cysts diagnosed in the intrauterine period. Focusing on the articles published in the last 10 years in the specialized literature, we tried to identify a conceptualization regarding the surveillance and treatment of these anomalies. The secondary objectives are represented by the assessment of the gestational age at which these anomalies appear, their distribution according to type (simple vs. complex), the assessment of the incidence of complications, and the assessment of ovarian tissue loss.

Scoping review questions:◦What are the results of conservative and surgical treatment in the case of fetal ovarian cysts?◦What is the gestational age at which fetal ovarian cysts are diagnosed?◦Of the diagnosed fetal ovarian cysts, what is the proportion of simple ovarian cysts and of complex ones?◦What is the risk of complications in the case of fetal ovarian cysts?◦In the case of surgical intervention, what are the results regarding ovarian preservation in the case of fetal ovarian cysts?

On the basis of the working protocol established and the main objective set, a series of keywords were established to identify the studies in the literature that would answer the questions formulated in this scoping review. The search was carried out by 2 authors in the PubMed, Embase, and Web of Science databases on 1 June 2022 using these keywords and MesH terms. Moreover, in order to compare the results, a search was carried out with the same criteria in the Google Scholar search engine. The first 50 results ranked as relevant were analyzed and compared to complete the information sources. No publications were found that met the eligibility criteria imposed and that did not represent a duplicate of the selected articles from the 3 specified databases.

To carry out the study, the following eligibility criteria were applied:-Articles published in the last 10 years (from 1 January 2012 to 1 June 2022).-Articles published in English.-Case series publications to ensure homogeneity. Case report type articles, review type articles were excluded (to avoid cases overlap).

Following the searches in the database, a number of 43 studies were identified that were subjected to further analysis. For an easy systematization, the PRISMA diagram (Preferred Reporting Items for Systematic Reviews and Meta-Analysis) was drawn up (Figure 1) [12]. Therefore, 43 case series publications with the evolution and management of fetal ovarian cysts were published in the last 10 years in the 3 databases, and after the elimination of duplicates, 42 studies remained (1 duplicate). Following the analysis of the type of study and screening of titles and abstracts, 20 articles were eliminated (case report type articles = 16, respective systematic review = 4).

The eligibility criteria were applied to the remaining articles, with 7 articles not fulfilling these criteria because 2 of the articles were in a language other than English, 1 article represented a national guide, 2 articles described only the possible surgical techniques, and 1 article was a review of a publication. Moreover, one article was eliminated at the time of data extraction because the subjects included were only cases with postnatal complications and whose initial diagnosis was other than ovarian cyst, thus missing data on diagnosis, size, treatment method, and evolution with reference to the pathology of interest of this scoping review.

## 3. Results

The 15 articles included for the analysis summarized a number of 517 girls who were identified with an ovarian cyst during the intrauterine period. Of the total, 14 studies were retrospective and were based on data collected from health unit registers and one article, Diguisto et al., that was prospective randomized (Table 1).

The conservative treatment method summarized 325 cases, of which in 85% (*n* = 273) of the cases it was successful, not requiring a surgical intervention in the follow-up specific to each study as a result of the disappearance of the ovarian cysts or their regression. Spontaneous regression of ovarian cysts was found more frequently in the case of simple ovarian cysts than in complex ones (69.23% vs. 30.77%, CI 95%). Intrauterine cyst aspiration was applied to 54 girls; in most cases, there was no feedback regarding the success rate with this treatment method.

Surgical treatment was necessary in 184 cases. From the surgical interventions performed, in 44.02% (*n* = 81) of the cases, it was possible to save the ovarian tissue, with the surgical techniques being represented by ovarian cystectomy or cyst aspiration. In more than half of the cases, 55.98% (*n* = 91), preservation of the ovarian tissue was not possible, which is why the surgical technique performed was oophorectomy or salpingo-oophorectomy. Here, we mention the fact that complicated cases with self-amputation of the affected ovary as a result of a chronic torsion were also included. These cases were identified intraoperatively by the lack of identification of the ovarian tissue at the level of the affected adnexa, with these cases representing 30.76% of the total surgical interventions (Table 2).

Regarding the risk of complications, the rate of their occurrence was analyzed in the included studies depending on the simple or complex type of the ovarian cyst, as shown in Figure 2.

Three studies were excluded from the analysis—Diguisto et al., Noia et al., and Ozcan et al.—because they only included cases with simple or complex ovarian cysts, and therefore the data were unable to be estimated. To analyze the data from the other 12 included studies, the “risk difference” was calculated using the Review Manager 5.1 software, the Mantel–Haenszel statistical method with the “fixed effect” analysis model, with the confidence interval being 95%. The analysis showed a 34% higher risk of complications in the presence of complex-type ovarian cysts. The result is statistically relevant, having a value of Z = 6.98, and *p* < 0.00001. A remark of this analysis is the increased heterogeneity, I2 = 75%, caused by the significant differences in the incidence of complications in the included studies.

## 4. Discussions

There is no algorithm regarding the treatment of fetal ovarian cysts. Most authors agree and propose monitoring it if the maximum diameter is below 40 mm and there are no complications or symptoms. In the case of masses over 40 mm, or which present symptoms or complications, early surgical treatment is indicated in order to save the ovarian tissue. In reality, the decision is always individualized on the basis of interdisciplinary consultations between the obstetrician, the neonatologist, and the pediatric surgeon. These consultations establish the risks and benefits from a medical point of view that need to be explained in detail with the couple in question, which is the final decision-making element. We consider family integration essential in decision making, both in the intrauterine and postpartum periods.

Another controversy that arose after the detection of the ovarian cyst in the intrauterine period was the choice among the options for childbirth. In the included studies, these choices did not vary according to the presence or absence of the fetal ovarian cyst. Most of the fetuses were born vaginally, with the exception of cases with obstetrical indication for caesarean section. There were no intrapartum complications for vaginal or caesarean section. Hara et al. took into consideration the induction of vaginal birth between weeks 37–39 of gestation, comparing the results with fetuses born after 39 weeks of gestation. Their work described similar rates of torsion in the two groups, concluding that induction of labor before 39 weeks has no advantage.

The minimally invasive treatment of the ovarian cyst includes intrauterine aspiration through ultrasound-guided puncture. The indications and results for this type of treatment are contradictory due to the complication rate. Among the studies of Cho et al., Diguisto et al., Hara et al., Noia et al., and Tyraskis et al. include cases of fetuses treated by this method. As we anticipated, the results are contradictory: Cho et al. describe a case of intrauterine aspiration that presented recurrence; Diguisto et al. describe a recurrence rate that required surgical intervention in 21% of cases; Hara et al. included four cases of intrauterine aspiration, two of which presented postpartum recurrence that required surgical intervention; and Tyraskis et al. included two cases of aspiration through intrauterine puncture and two cases of aspiration through postpartum puncture, one of which was performed intrauterine and presented cyst recurrence.

## 5. Conclusions

The fetal ovarian cyst is the most frequently intra-abdominal mass diagnosed in the intrauterine period. The causes that lead to the appearance of this anomaly are incompletely elucidated but they seem to be related to the exposure of the ovarian tissue to increased levels of maternal hormones and the hormones secreted by the placental tissue. The diagnosis is established by ultrasound during the routine monitoring of the pregnancy. Ultrasound evaluation allows for the classification into simple ovarian cyst and complex ovarian cyst. The type of cyst in accordance with its size are the main criteria according to which the therapeutic approach is decided. In the absence of complications and clinical symptoms at birth, the monitoring of cystic masses can lead to their regression and even their disappearance over time. The success rate is higher in the case of simple cystic masses and is inversely proportional to their size. Complex cystic masses, especially if their size is over 40 mm, there is the presence of clinical symptoms at birth, and there is the presence of complications, require surgical treatment. The applied surgical techniques aim to preserve the ovarian tissue and remove the cystic mass. However, in a significant percentage of cases, saving the ovarian tissue is impossible either through necrosis produced by the torsion of the cystic mass or by self-amputation of the ovary through chronic torsion, in which case, no ovarian tissue is found at the level of the respective adnexa.

## Figures and Tables

**Figure 1 medicina-59-00186-f001:**
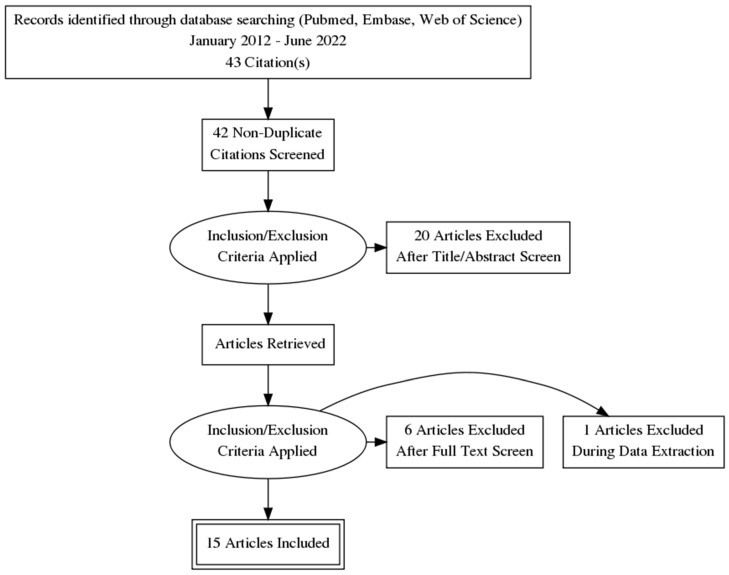
PRISMA flow diagram illustrating the search strategy.

**Figure 2 medicina-59-00186-f002:**
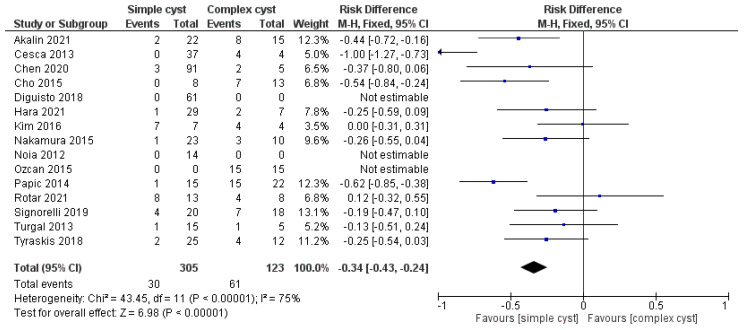
Forest plot comparing the risk of complication in simple cysts vs. complex cysts.

**Table 1 medicina-59-00186-t001:** General characteristics of studies.

	Nr. Fetuses	Gestational Age at Diagnosis (WG)	Mother Age (Years)	Type of Cyst—Simple	Type of Cyst—Complex	Dimension (mm)
[7] Tyraskis et al., 2018	37	NS	NS	25	12	NS
[13] Cesca et al., 2012	41	32.2	34.5	37	4	38.8
[14] Kim et al., 2016	11	NS	35	7	4	53
[15] Hara et al., 2021	36	32	30	29	7	32
[16] Rotar et al., 2021	21	31.28	29.95	13	8	NS
[17] Ozcan et al., 2015	15	32.9	NS	0	15	51.8
[18] Diguisto et al., 2018	61	33.5	28.5	61	0	42
[19] Noia et al., 2012	14	32.25	32.84	14	0	48.06
[20] Papic et al., 2014	37	NS	NS	15	22	53.48
[21] Akaline et al., 2021	37	31.62	28.02	22	15	42.17
[22] Turgul et al., 2013	20	28.4	26.4	15	5	40.89
[23] Cho et al.,2015	21	NS	NS	8	13	50
[24] Nakamura et al., 2015	31	32	32	21	10	NS
[25] Chen et al., 2020	96	32	28	91	5	30.2
[26] Signorelli et al., 2019	39	NS	NS	20	19	33.05
Total/average	517	31.82	30.52	378	139	42.95

NS = not stated; WG = week of gestation.

**Table 2 medicina-59-00186-t002:** Treatment applied for fetal ovarian cyst.

	Conservative—Watch and Wait—Total	Watch and Wait—Regression	Intrauterine Aspiration	Total Surgery	Ovarian Preservation	Radical Surgery
[7] Tyraskis et al., 2018	26	26	2	8	1	7
[13] Cesca et al., 2012	37	37	0	4	0	4
[14] Kim et al., 2016	11	2	0	9	1	8
[15] Hara et al., 2021	23	23	4	13	10	3
[16] Rotar et al., 2021	6	0	0	15	5	10
[17] Ozcan et al., 2015	0	0	0	15	0	15
[18] Diguisto et al., 2018	27	17	34	17	9	8
[19] Noia et al., 2012	0	0	14	3	0	3
[20] Papic et al., 2014	25	12	0	25	14	11
[21] Akaline et al., 2021	24	24	0	13	6	7
[22] Turgul et al., 2013	20	18	0	2	0	2
[23] Cho et al.,2015	0	0	0	21	15	6
[24] Nakamura et al., 2015	17	17	0	14	11	3
[25] Chen et al., 2020	83	71	0	13	9	4
[26] Signorelli et al., 2019	26	26	0	12	NS	NS
Total/average	325	273	54	184	81	91

NS = not stated.

## Data Availability

Not applicable.

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
