# Peer review of "Fetal Ovarian Cyst—A Scoping Review of the Data from the Last 10 Years"

_medicina, 2023, doi:10.3390/medicina59020186_

Round 1
Reviewer 1 Report
The paper is well written e provides a wide and comprehensive overview of the argument. Though there aren't so many new findings, it can be useful to sum up knowledge on the topic and to give the clinicians elements to improve their counselling. The review of the literature is complete and exhaustive. The case report is interesting and well described. Conclusions are balanced and useful.
Author Response
Hello.
I attach here the article with the suggested changes.
Best regards,
Dr. Ionel Nati

Reviewer 2 Report
Despite the low prevalence, the optimal clinical management and outcome of fetal ovarian cysts diagnosed by routine ultrasound screening is relatively intensively discussed in the literature, however there are open questions in the daily practice and the treatments are usually individual and based on personal experience. Thus, any comprehensive review may add a little piece to the clinical guidance. This review aims to discuss the results of the conservative and surgical treatments, second point are the gestational age at diagnosis, proportion of simple/complex cysts and complication, with the rigorous analysis of 15 case-series papers. A case report is added to illustrate one of the possible clinical scenarios. The topic is important, I read it with interest.
In general:
First, although the paper is well written, the text is too extensive and many well known information is discussed abundantly. The whole paper should be shortened.
Second, after the review of Bascietto et ail in 2017 based on more patient (Outcome of fetal ovarian cysts diagnosed on prenatal ultrasound examination: systematic review and meta-analysis. Ultrasound Obstet Gynecol. 2017 Jul;50(1):20-31.) the results barely add new information. However, since the rate of the chronic transition is similarly high (ca. 10%), the results favor the early postnatal surgical management of not only the complex, but the simple cyst over 4-5 cm diameter to preserve ovarian tissue. This massage can be strenghten even more.
Third, the case its the treatment presented is not unique and does not illustrate the complexity of the question. It may be put into the introduction as explaining the trigger of the literature search.
Some comments, which affect clinical aspects:
Since this is not only a review, but an overview of the fetal ovarian cyst, more illustrations (ultrasound and surgical) would be interesting for the readers to see the spectrum of the disease.
I really miss the discussion of the preferred mode of the delivery depending on the size, type and the growing dynamics of the cysts. What is the rate of the rupture/torsion during vaginal birth versus caesarian, and does it cause any harm?
Regarding the complications, with intrauterine puncture what is the rate of the abdominal bleeding or recurrence.
Very important issue is the interdisciplinary consultation (prenatal specialist, neonatologist, pediatric surgeon) involving the parents into the decision. The authors underline its significance in the postnatal situation, but to my opinion it has to be done also prenatally.
Overall, my impression that there are extensive efforts put into this paper, but it still needs extensive revision in order to provide new practical information for the clinicians.
Author Response

(The authors gave the same response as above.)

Round 2
Reviewer 2 Report
Thank you for supplementing the discussion according to my suggestions, I feel these improved the value of this review.